# Resonant Excitation of the Ferroelectric Soft Mode by a Narrow-Band THz Pulse

**DOI:** 10.3390/nano13131961

**Published:** 2023-06-28

**Authors:** Kirill Brekhov, Vladislav Bilyk, Andrey Ovchinnikov, Oleg Chefonov, Vladimir Mukhortov, Elena Mishina

**Affiliations:** 1Department of Nanoelectronics, MIREA—Russian Technological University, Moscow 119454, Russia; brekhov_ka@mail.ru (K.B.); mishina_elena57@mail.ru (E.M.); 2Institute for Molecules and Materials, Radboud University, 6525 AJ Nijmegen, The Netherlands; 3Joint Institute for High Temperatures of Russian Academy of Sciences (JIHT), Moscow 125412, Russia; a.ovtch@gmail.com (A.O.);; 4Southern Scientific Center of Russian Academy of Sciences, Rostov-on-Don 344006, Russia; mukhortov1944@mail.ru

**Keywords:** THz excitation, ferroelectric, second harmonic generation

## Abstract

This study investigates the impact of narrow-band terahertz pulses on the ferroelectric order parameter in Ba_0.8_Sr_0.2_TiO_3_ films on various substrates. THz radiation in the range of 1–2 THz with the pulse width of about 0.15 THz was separated from a broadband pulse with the interference technique. The 375 nm thick BST film on a MgO (001) substrate exhibits enhanced THz-induced second harmonic generation when excited by THz pulses with a central frequency of 1.6 THz, due to the resonant excitation of the soft phonon mode. Conversely, the BST film on a Si (001) substrate shows no enhancement, due to its polycrystalline state. The 800 nm thick BST film on a MgO (111) substrate demonstrates the maximum of a second harmonic generation signal when excited by THz pulses at 1.8 THz, which is close to the soft mode frequency for the (111) orientation. Notably, the frequency spectrum of the BST/MgO (111) film reveals peaks at both the fundamental and doubled frequencies, and their intensities depend, respectively, linearly and quadratically on the THz pulse electric field strength.

## 1. Introduction

Today, an important task is to develop methods for the ultrafast control of the ferroelectric order parameter. Solving this problem will allow for creating new energy-efficient optoelectronic devices such as memory cells and modulators [1,2,3,4]. In magnetic materials, this problem has already been solved using optical [5,6] or terahertz [7,8,9] pulses. However, in the case of ferroelectrics, this problem remains unsolved, since the mechanisms that allow to achieve ultrafast switching remain unclear [10,11,12,13].

At present, there are some theoretical [14,15,16,17,18] and experimental works that show the possibility of influencing the ferroelectric order parameter with optical [19,20,21] or THz pulses. The THz-induced change in polarization can be directly measured as a dynamical polar ion displacement with time-resolved X-ray diffraction [22]. Such an effect was demonstrated in Sn_2_P_2_S_6_ ferroelectric crystal in [23] and incipient ferroelectric SrTiO_3_ [24]. More often, the THz-induced change in polarization is probed optically, using second harmonic generation (SHG) or the optical Kerr effect. Using these effects, transient polarization switching or ultrafast polarization modulation was demonstrated in SrTiO_3_ [25], an organic molecular dielectric [26], ferroelectric crystal BaTiO_3_ [27], and PGO [28]. Recently, THz pulse excitation was shown to generate neuronic and synaptic behavior in relaxor ferroelectric Pb(Mg_1/3_Nb_2/3_)O_3_ (PMN) due to the activation of hidden phases [29], which can be considered as a type of ultrafast polarization switching in this material. Almost all experiments have been performed using a single period, i.e., a broadband THz pulse. The polar mode that is considered as a driving force for switching is excited non-resonantly in this way.

Several theoretical works [18,30] predicted a higher impact on polarization with the direct excitation of higher-frequency modes (IR or Raman) rather than the low-frequency soft mode; the process is followed by the excitation of the soft mode due to the phonon–phonon interaction. The experiment was realized in LiNbO_3_ [30] with the resonant excitation of the *Q_IR_* mode with the frequency of 19 THz, which led to polarization modulation detected by the SHG technique. Experiments on the direct excitation of the soft mode have not yet been carried out due to the lack of suitable THz sources.

Ba_0.8_Sr_0.2_TiO_3_ (BST) thin films have already proven themselves as materials that respond efficiently to broadband terahertz pulses [31,32,33]. Additionally, these materials are radiation-resistant, energy-efficient, and possess a low dielectric loss and stable parameters in a wide temperature range [34]. BST is a uniaxial ferroelectric with a perovskite structure. At room temperature, BST is in the ferroelectric phase with a tetragonal unit cell (space group P4mm) and goes to the paraelectric phase with a cubic unit cell (space group Pm3m) above the Curie temperature of 353 K. The soft-mode frequency can be varied by changing the crystallographic face, chemical composition, and substrate material.

Magnesium oxide is a typical substrate material for BST films and other perovskite-like ferroelectrics, since they have fairly close cell parameters, and there is a well-established deposition technology, which makes it possible to obtain high-quality films. However, the integration of ferroelectric films into silicon technology is currently a topical issue [35]. For instance, the use of ferroelectrics as a gate in a field-effect transistor makes it possible to create non-volatile and electrically reprogrammable memory cells [36]. So far, not all problems have been solved. Ferroelectric materials are deposited on silicon substrates at high temperatures. This leads to the mutual diffusion of the components at the ferroelectric–semiconductor interface, the formation of a transition layer, and the deterioration of the crystal structure [37]. As a result, an instability of the polarized state occurs and the number of rewriting cycles is reduced. To solve these problems, transition layers based on simple oxides such as (Ba,Sr)TiO_3_ [38,39] with a thickness of several nanometers are formed at the interface between the semiconductor and the ferroelectric [40]. In addition, depositing BST on silicon when creating solar cells enables a higher absorption of visible and ultraviolet radiation [41,42], which significantly increases their efficiency.

Here, we present the results of the investigation of the resonant excitation of the soft mode in Ba_0.8_Sr_0.2_TiO_3_ ferroelectric films deposited on MgO(001), MgO(111), and Si(001) substrates by strong narrow-band THz pulses in the range of 1–2 THz.

## 2. Materials and Methods

We investigated the nonlinear optical response dynamics in three barium–strontium titanate (Ba_0.8_Sr_0.2_TiO_3_, BST) films under excitation by narrow-band THz pulses of a few-picosecond duration. These samples were fabricated with radio-frequency (RF) sputtering of a Ba_0.8_Sr_0.2_TiO_3_ ceramic target. The first sample was a 375 nm thick BST film deposited on a (001)-oriented MgO substrate. The second sample had the same thickness and composition, but with a sublayer of the same film with a thickness of 2.5 nm and deposited on a silicon substrate. The substrate was the single crystal p-type silicon with a resistivity of 12 Ohm/cm and crystallographic orientation (001) [42]. The third sample was an 800 nm thick BST film on a (111)-oriented MgO substrate. The sample fabrication processes are described in more detail in [35,40,42]. The film thickness was measured using an MII-4 microinterferometer and also a Zeiss Supra-25 scanning electron microscope in the “cross-section” and AFM image of the end face of the etched film. From the height of the resulting step, the BST film thickness was determined.

The XRD patterns of BST/MgO(001), BST/Si(001), and BST/MgO (111) samples are shown in Figure 1a–c, respectively.

Figure 1a shows that the 375 nm thick BST film on the MgO (001) substrate is a single crystal, with the spontaneous polarization vector directed perpendicularly to the substrate surface. The lattice parameters for this sample are *c* = 0.4041 ± 0.0003 nm and *a* = 0.3960 ± 0.0003 nm. In contrast, Figure 1b demonstrates that the 375 nm thick BST film on the (001)-oriented silicon substrate is polycrystalline, containing crystallites with (001) and (011) plane orientations parallel to the substrate plane. The spontaneous polarization vector for (001) crystallites is perpendicular to the substrate surface, while for (011) crystallites, it is at an angle of about 45 degrees. The lattice parameter for this film is *c* = 0.4035 ± 0.0003 nm.

Figure 1c shows that the BST film on the MgO(111) substrate is epitaxial and contains only a rhombohedral phase, with polarization directed perpendicularly to the substrate. The unit cell parameters are *a* = 0.39616 ± 0.0005 nm and *α* = 89.519°. The XRD pattern demonstrates that the film is oriented along the [111] direction, so [111]_film_ || [111]_MgO_. The unit cell volume of the rhombohedral BST film is 1.5% less than that of the tetragonal ceramic target.

It should be noted that the BST/MgO(001) film has strong stress at the film/substrate interface, significantly higher than in the BST film on the (111)MgO substrate. In the silicon film, this stress is also considerably lower due to the 2.5 nm thick BST sublayer. This is due to discontinuous deposition with cooling to room temperature of the buffer layer before the deposition of the film [43]. This makes it possible to separate the nucleation process from the growth process, which leads to a decrease in stresses and promotes the epitaxial growth of BST in the ferroelectric phase [35].

It also should be noted that all three samples were depolarized before the acting of narrow-band THz pulses, according to [34,44,45].

The dynamics of a nonlinear optical response in BST films when they are excited by narrow-band THz pulses were investigated using a modified experimental setup described in [32].

A Cr:forsterite laser system with a wavelength of 1240 nm, a pulse repetition rate of 10 Hz, and a duration of 100 fs was used to generate narrow-band THz pulses. THz pulse generation occurred in the organic crystal OH1. To generate narrow-band THz pulses, the amplified laser pulse was split into two parts, with each passing through one arm of a Mach–Zehnder-type interferometer. By adjusting the delay between these pulses before compression, it was possible to achieve the beating of the optical pulses at a necessary frequency [46]. The resulting frequency-modulated optical chirp irradiated the OH1 crystal, generating narrow-band terahertz radiation. The spectral line width was 0.1–0.2 THz, depending on the central frequency. Measurements of the temporal waveform of narrow-band THz radiation pulses were performed in the electro-optical detection scheme on a 1 mm thick ZnTe crystal [47]. Figure 2 shows (a) the temporal waveform and (b) the corresponding spectrum of narrow-band THz pulses at a frequency of 1.2 THz.

Similar measurements of temporal waveforms were made in a frequency range of 1.2 to 2.0 THz. To estimate the electric field strength of THz pulses at various frequencies, we measured the energy of THz pulses and the spatial distribution of the THz beam in the focal plane of the focusing off-axis parabolic mirror. THz pulse energy measurements were made using a Golay cell, while the spatial distribution was measured with a terahertz camera. Thus, we obtained a comprehensive set of experimental data on the terahertz source parameters, enabling us to estimate the electric field strength at different frequencies. The electric field range was estimated according to [37] as 320 kV/cm–1.45 MV/cm, depending on the central generation frequency.

Detection occurred at the frequency of the second optical harmonic. This technique is one of the most sensitive methods to study the order parameter of ferroelectrics [38,39,40]. The BST films deposited on the MgO substrate were measured in transmission geometry with a normal and 45-degree incidence, respectively. The difference in geometry led to different values of the coherence length, which is discussed in Section 3.

Table 1 compares the samples as well as the experimental geometries.

## 3. Results

Figure 3a–c demonstrates the dependence of the normalized THz-induced second harmonic signal dynamics on the time delay between pump and probe pulses in BST/MgO (001), BST/Si (001), and BST/MgO (111) films, respectively, when the narrow-band pump pulses were in the range of 1.2–2.0 THz. The pump frequencies were chosen for each sample based on the proximity to the soft mode for its most effective excitation.

As can be seen from Figure 3a–c, the intensity of the THz-induced SHG signal for all three samples correlates with the temporal waveform of the excitation THz pulse, shown in Figure 2a. It should be noted that the THz-induced SHG signal depends on the pump frequency for various samples in different ways. For example, for the BST/MgO (001) film, the highest SHG intensity is observed at the excitation THz pulse frequency of 1.6 THz (Figure 3a). In Figure 3b,c, it is not possible to unambiguously identify the pumping frequency that provides the highest SHG signal.

Figure 4a–c demonstrates the frequency spectra of the dependences shown in Figure 3, obtained with the Fourier transform in the region up to 2.5 THz. For comparison, the green dashed lines represent the spectrum of the SHG signal measured when the samples are excited by broadband THz pulses with an electric field up to several MV/cm [48].

It should be noted that for a correct comparison, the obtained frequency dependences were successively normalized to the probe power, film thickness with regard to the coherence length, and electric field of THz radiation. When performing FFT, we limited the consideration of temporal dependences by 17.5 ps (as in Figure 3) to avoid the distortion of the spectra by the reflection of THz pulses from the backside of the substrate as well as other effects of the pulse propagation [32]. The coherence length was estimated using the expression *l_coh_* = *λ*_ω_/4(*n*_ω_ ± *n*_2ω_), where *n*_ω_ and *n*_2ω_ are the refractive indices for the fundamental and SHG waves, and + (−) refers to the reflection (transmission) geometry, respectively. Thus, for BST films on MgO substrates, the coherence length was found as 1630 ± 90 nm and 1350 ± 90 nm for MgO(001) and MgO(111) substrates, respectively. This value exceeds the film thickness, allowing us to use the exact sample thickness for normalization. For the BST film on the Si substrate, the coherence length was estimated as 67 ± 2 nm. This value was used in the normalization.

Normalization on the THz field was performed relative to the first degree of its strength. This is due to the fact that in the presented frequency spectra, only the peaks corresponding to the frequency of excitation THz pulses are observed. All the obtained dependences were normalized to the square of the optical probe radiation power.

Figure 4a demonstrates that the spectral amplitude of the signal measured at the excitation pulse frequency of 1.6 THz significantly exceeds the spectral amplitudes of the signals obtained at other excitation pulse frequencies. This can be explained as follows. The soft ferroelectric mode in the BST film on the MgO (001) substrate has a frequency of about 1.67 THz [31,32,49]. By exciting this material with a narrow-band THz pulse of a suitable central frequency, we act on the polar ion more effectively, displacing it from the equilibrium state, thereby increasing the polarization of the medium. Since the SHG signal intensity is proportional to the square of the polarization, the amplification of the SHG signal at 1.6 THz shown in Figure 4a may be explained by the resonance effect on the polar ion.

Figure 4b shows that for the BST/Si (001) film, the spectral amplitudes are not significantly different, while the amplitude for the 2.0 THz pump frequency, it is considerably smaller. It is important to note that in this case, there is no pronounced resonance amplification of the signal. This can be explained by the fact that the sample BST/Si (001) is polycrystalline with regions with different polarization directions relative to the sample surface, which are not affected equally by excitation THz pulses.

Figure 4c demonstrates the frequency spectrum for the BST/MgO (111) film in the range up to 2.5 THz. It can be seen that the peak corresponding to the excitation pulse frequency of 1.8 THz has the highest intensity. This can be explained by the fact that according to [50], the soft ferroelectric mode in Ba_0.8_Sr_0.2_TiO_3_ film on a MgO substrate with a crystallographic orientation (111) has a frequency of about 1.9 THz. The maximum amplitude of the peak with a frequency of 1.8 THz, rather than 1.9 THz, can be explained by the width of the spectral line in this range of about 0.2 THz.

Thus, Figure 4a,c demonstrate that the use of narrow-band THz pulses to excite phonon modes is much more efficient than broadband THz pulses. In particular, they make possible the resonant excitation of the soft phonon mode in ferroelectrics.

Figure 5 presents the frequency spectra for the BST/MgO (111) film in the range from 2.2 THz to 4.5 THz. For this sample, the frequency spectra show a second peak at the double frequency of the excitation THz pulse. No second peak is observed in the frequency spectra of the other two samples. These dependences were normalized to the square of the THz field strength.

The presence of two peaks in the spectrum at the fundamental and at the doubled frequency can be explained by the fact that the dependence of the SHG intensity on the external electric field is quadratic. Indeed, the SHG intensity in the THz field can be represented as a decomposition either by the THz field *E*_Ω_ in the case of a non-ferroelectric crystal,
(1)$$I^{2\omega }(E_{\Omega })\propto {\left(\chi^{\left(2\right)}+\chi_{E}^{\left(3\right)}E_{\Omega }\right)}^{2}{\left(I^{\omega }\right)}^{2},$$
or by the polarization *P*(*E*_Ω_) in the case of a ferroelectric crystal,
(2)$$I^{2\omega }(P(E_{\Omega }))\propto {\left(P_{0}+\chi_{P}^{\left(3\right)}P(E_{\Omega })\right)}^{2}{\left(I^{\omega }\right)}^{2},$$
where *χ*^(2)^(2ω; ω, ω)—crystallographic quadratic susceptibility and *χ*^(3)^*_E_*(2ω; Ω, ω, ω)—cubic susceptibility. The cubic susceptibility can be considered as a measure of polarization switchability—the higher the value of *χ*^(3)^, the lower THz field is required to control (switch) polarization.

Obviously, in the case of a linear dependence of *P*(*E*_Ω_), for example, in weak fields, relations (1) and (2) are identical. In the general case, in order to distinguish (1) and (2), it is necessary to investigate the dependences of the SHG intensity on the THz field.

When decomposing the square of the sum, two field-dependent terms appear—linear *I*_2_ and quadratic *I*_3_:(3a)$$I_{2}\propto \chi^{\left(2\right)}\chi_{E}^{\left(3\right)}E_{\Omega },$$
(3b)$$I_{3}\propto {\left(\chi_{E}^{\left(3\right)}\right)}^{2}{\left(E_{\Omega }\right)}^{2}.\;$$

These terms in the Fourier decomposition give, respectively, signals at the fundamental Ω and doubled 2Ω frequencies of the incident wave.

A comparison of the frequency amplitudes of the optical SHG signal at the first terahertz harmonic (Ω) for three samples shows the highest system excitation efficiency for the BST/MgO (111) structure. Figure 5 also confirms this statement—only BST/MgO (111) has a signal at the second terahertz harmonic (2 Ω). This is quite expected, since there is a component in the polarization that coincides with the direction of the terahertz field (in the film plane). On the other hand, three types of domains oriented at 120 degrees relative to each other can reduce the signal [51]. However, when considering terahertz excitation, we take into account the crystallography of the unit cell. According to all available data, the characteristic time of domain wall motion is less than the period of a terahertz pulse.

However, the complex domain structure may provide an additional pathway for switching through the interaction of ferroelastic domains. It was recently shown in [52], on a 70 nm thick PbTiO3 film, that tuning the system to the brink of structural instability by controlling epitaxial deformation makes it possible to switch collective domains. The effect has been demonstrated by applying a local mechanical force with the cantilever of a piezoresponse force microscope, but it appears to be non-local. We believe that structural instability may increase excitation efficiency, but this effect should be carefully studied.

It is important to emphasize that both the experiment and simulation show only phonon excitation, which corresponds to dynamic switching, that is, the collective motion of a polar ion within a terahertz pulse width. In our experiments, a sequence of identical pulses was used, and the signal was averaged over the sequence. No permanent switching was observed after the end of the pulses. A possible reason has been suggested in [18]. It is stated that a single-pulse excitation of a sufficient intensity will always lead to sample depolarization due to the chaotic dynamics of a system of coupled harmonic oscillators with a symmetric potential well. In order to obtain permanent switching, the authors of [18] proposed either to break the symmetry of the potential energy surface by an additional interaction (electric or magnetic field, anisotropic voltage, etc.), or to use a special sequence of pulses, which was first proposed in [53].

## 4. Conclusions

In conclusion, we demonstrated two important issues of the developed technique. Firstly, the narrow-band THz pulse may excite the specific modes in a ferroelectric, including the soft mode. This should result in dynamical polarization switching, analogously to the case of broadband excitation. Secondly, excitation is much stronger for frequencies close to a specific mode, which means resonant excitation. This is in contrast to broadband excitation, where the resonances cannot be isolated.

The developed narrow-band THz spectroscopy allows us to compare the impact of the input THz field on ferroelectric polarization measured by SHG in different samples. The highest impact or the highest switchability was achieved in 800 nm thick BST film on an MgO (111) substrate. This can be explained by the presence in this sample of a polarization component parallel to the THz field. The lowest impact or the lowest switchability was obtained in 375 nm thick BST film on a Si (001) substrate. This is due to its polycrystalline structure. The single crystalline BST film on an MgO (001) substrate demonstrates an intermediate impact.

The experiments were performed using the unique scientific facility “Terawatt Femtosecond Laser Complex” in the “Femtosecond Laser Complex” Center of the Joint Institute for High Temperatures of the Russian Academy of Sciences.

## Figures and Tables

**Figure 1 nanomaterials-13-01961-f001:**
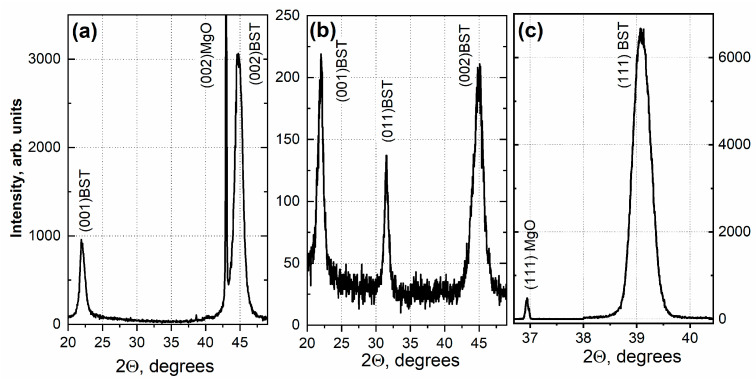
XRD patterns of (**a**) BST/MgO (001), (**b**) BST/Si (001), (**c**) BST/MgO (111).

**Figure 2 nanomaterials-13-01961-f002:**
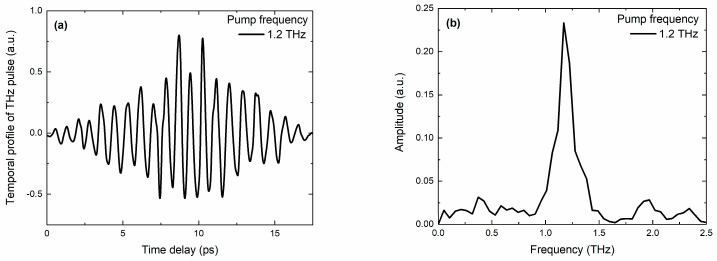
(**a**) Temporal waveform and (**b**) spectrum of narrow-band THz pulses at 1.6 THz.

**Figure 3 nanomaterials-13-01961-f003:**
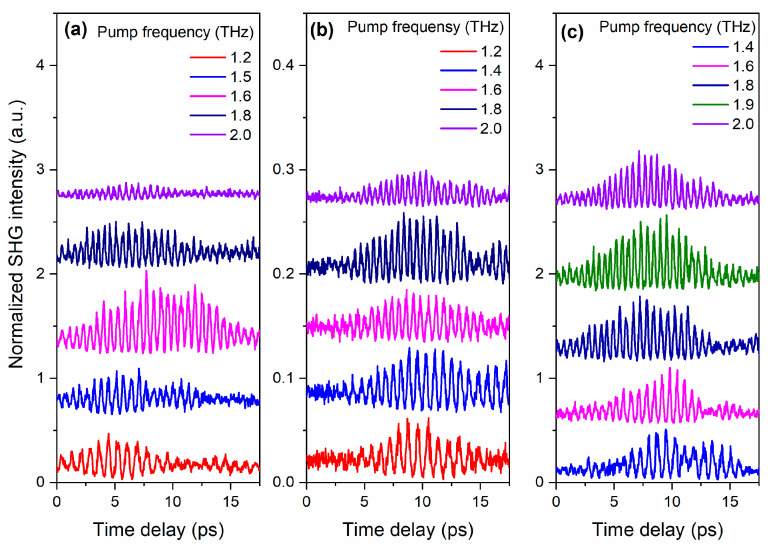
Dynamics of the normalized THz-induced SHG intensity for (**a**) BST/MgO (001), (**b**) BST/Si (001), (**c**) BST/MgO (111) at different central frequency of the excitation THz pulse. For the convenience of observation, the plots are shifted along the ordinate axis.

**Figure 4 nanomaterials-13-01961-f004:**
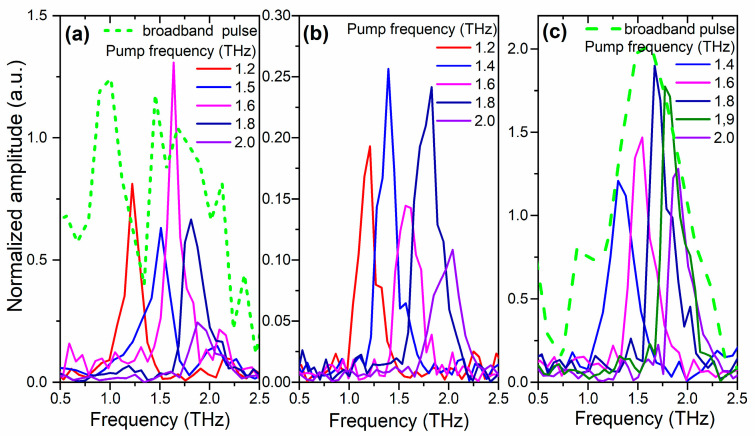
Frequency spectra of the dependences presented in Figure 3 for films of (**a**) BST/MgO (001), (**b**) BST/Si (001), (**c**) BST/MgO (111), respectively. The green dashed line shows the SHG frequency spectrum obtained from the samples when they were excited by a broadband THz pulse.

**Figure 5 nanomaterials-13-01961-f005:**
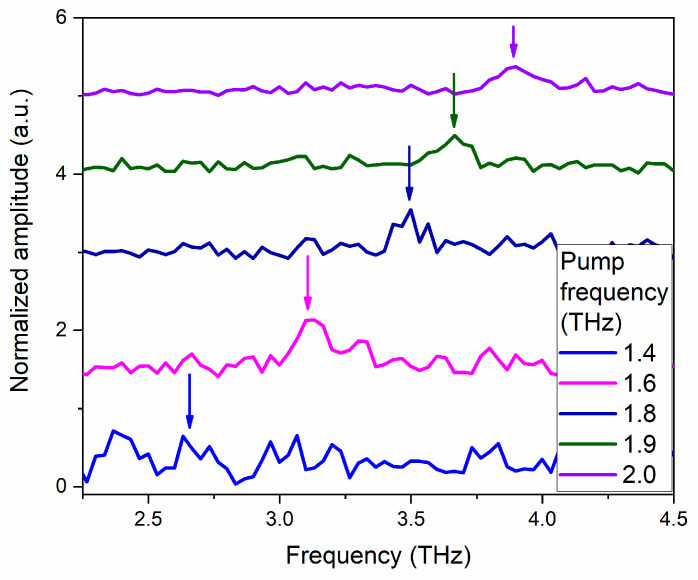
Frequency spectra for the BST/MgO (111) sample in the range of 2.2–4.5 THz. The normalized amplitude on Figure 5 is three orders of magnitude less than on Figure 4.

**Table 1 nanomaterials-13-01961-t001:** The samples.

	Ba_0.8_Sr_0.2_TiO_3_/ MgO(001)	Ba_0.8_Sr_0.2_TiO_3_/ Si(001)	Ba_0.8_Sr_0.2_TiO_3_/ MgO(111)
Crystal structure	Single crystal	Polycrystalline	Single crystal
Crystallographic orientation	(001)	(001, 011)	(111)
Thickness d, nm	375	375	800
Geometry	Transmission, 0°	Reflection, 45°	Transmission, 0°
	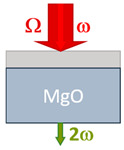	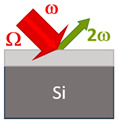	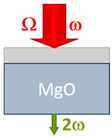
Coherence length, nm	1630 ± 90 nm	67 ± 2 nm	1350 ± 90 nm
THz field, kV/cm	700 ÷ 1400	700 ÷ 1400	320 ÷ 560

## Data Availability

The data presented in this study are available on request from the corresponding author.

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
