# Peer review of "Resonant Excitation of the Ferroelectric Soft Mode by a Narrow-Band THz Pulse"

_nanomaterials, 2023, doi:10.3390/nano13131961_

Round 1
Reviewer 1 Report
The author demonstrated the effects of narrow-band terahertz pulses on the polarization switching mechanism in Ba0.8Sr0.2TiO3 ferroelectric films. The resonant excitation of the soft phonon mode of different samples was observed. The manuscript can be accepted if the following issues are addressed:
1. Review of experimental results of terahertz pulses on the polarization switching mechanism should be enhanced. The boundary between the broad-band and narrow-band THz pulses should be emphasized.
2. The paper aims at the achievement of ultrafast polarization switching under narrow-band terahertz pulses, while the polarization states of three samples before and after the applying of narrow-band terahertz pulses are not given.
3. Figure 3(b-c) shows the similar SHG signals with different pump frequencies. More detail explanation should be given on this phenomenon. The significance of all pictures in Figure 3 needs to be explained in more detail.
4. The coherence length of <001> and <111> BST films for normalize the SHG intensity used in Table 1 is the same. Details should be given for the lattice orientation independence of out-of-plane refractive indices.
5. Some errors exist in figures, for example: B0,8Sr0,2TiO3
Author Response
Dear Reviewer,
Thank you for your time and valuable comments to improve our manuscript.
In response to your questions, we can answer the following:
- 1. Review of experimental results of terahertz pulses on the polarization switching mechanism should be enhanced. The boundary between the broad-band and narrow-band THz pulses should be emphasized.
The introduction was rewritten. The use of THz pulses as a pump to affect various ferroelectric materials has been described in more detail.
- The paper aims at the achievement of ultrafast polarization switching under narrow-band terahertz pulses, while the polarization states of three samples before and after the applying of narrow-band terahertz pulses are not given.
The co-author of our paper, V.Mukhorotov, is the owner of the manufacturing technology, he fabricated the samples and characterized them. All the properties correspond to the known ones: narrow ferroelectric hysteresis loops symmetrical with respect to the origin in the voltage-polarization axes and do not hold polarization for a long time (days). Examples of such loops are given in [DOI:10.1134/S1063783418090202, DOI:10.1134/S1063783418010250, DOI:10.1134/S106378421108024X]. This is also confirmed by microscopic images of samples with noise level SHG. This means that all experiments start with depolarized samples.
The following text was added to the manuscript (line 115):
« It also should be noted, that all three samples are depolarized before the action of narrow-band THz pulses, according to [34,43,44]»
- Figure 3(b-c) shows the similar SHG signals with different pump frequencies. More detail explanation should be given on this phenomenon. The significance of all pictures in Figure 3 needs to be explained in more detail.
Figure 3 has been revised from the previous version of the manuscript. Figure 3 (a-c) demonstrates the dependence of the normalized THz-induced second harmonic signal dynamics on the time delay between pump and probe pulses in various BST films, when the narrow-band pump pulses were in the range of 1.2 THz - 2.0 THz. The pump frequencies were chosen for each sample based on the proximity to the soft mode for its most effective excitation. The figure 3 (а-с) demonstrates that the amplitude of the THz-induced SHG signal depends on the pump frequency for various samples in different ways.
The text was added to the manuscript (line 159): “The pump frequencies were chosen for each sample based on the proximity to the soft mode for its most effective excitation.”
And (line 170) “It should be noted, that the THz-induced SHG signal depends on the pump frequency for various samples in different ways.”
- The coherence length of <001> and <111> BST films for normalize the SHG intensity used in Table 1 is the same. Details should be given for the lattice orientation independence of out-of-plane refractive indices.
Thanks for the important note! The coherence lengths were recalculated taking into account the crystallographic orientation of the samples. The changes have been made in Table 1 and in the text (lines 191-193).
- Some errors exist in figures, for example: B0,8Sr0,2TiO3
These typos have been fixed.
Reviewer 2 Report
The paper from Brekhov et al. discusses THz second harmonic generation from three Ba0.8Sr0.2TiO3 samples. According to the authors the signal increases when the excitation energy is resonant with the soft mode in the (001) crystal. Unfortunately I cannot share their view that the SHG signal increases close to a specific frequency. More in general I cannot distinguish any specific trend in their data, which makes their conclusions rather unsubstantiated.
Other more specific issues are the following:
1) The normalisation procedure employed to take into account the difference in the intensity of the pumping frequency is not explained
2) No prior characterisation of the THz properties of the materials have been done, so that the exact frequency of the soft mode is not known a priori
3) I cannot understand how the discussion about fig5 contributes in supporting the conclusions of the paper
4) The paper contains typos and mistakes, including in the formula of the material in which Ba is often misspelled with B
Author Response
Dear Reviewer,
Thank you for your time and valuable comments to improve our manuscript.
In response to your questions, we can answer the following:
1) The normalisation procedure employed to take into account the difference in the intensity of the pumping frequency is not explained.
For a correct comparison, the obtained dependences were normalized to the probe beam power, film thickness with regard to the coherence length, and the electric field of THz radiation.
SHG signal is proportional to the square of the optical pump power. Different optical powers were used for different samples. Due to this, the obtained dependences were normalized to the square of the optical radiation power. The SH signal is generated in a volume determined by the coherence length. Therefore, the coherence length of each film was calculated taking into account the geometry of the experiment and the crystallography of the sample. The SHG signal also depends on the exciting THz field. Therefore, the electric field strength for each pump frequency was calculated. Taking into account all these factors allow us to correctly compare the spectral amplitudes of peaks obtained by the Fourier transform and make conclusions about the resonant excitation of phonon modes.
The following text was added to the manuscript (line 198): “All the obtained dependences were normalized to the square of the optical probe radiation power”.
2) No prior characterisation of the THz properties of the materials have been done, so that the exact frequency of the soft mode is not known a priori.
You are right, the exact frequency of the soft mode is not known in advance. Moreover, the soft mode frequency can be varied by changing the crystallographic face, chemical composition, and substrate material. However, there are several works suggesting its approximate frequency in BST of different crystallographic orientations [Doi: 10.1209/0295-5075/112/47001, DOI: 10.1002/pssb.201600413, DOI: 10.1134/S1063783416100048]. We relied on these works when planning the experiment.
3) I cannot understand how the discussion about fig5 contributes in supporting the conclusions of the paper.
The results presented in Fig. 5 are not related to the efficiency of excitation of the soft mode in BST films when they are excited by narrow-band THz pulses. These are additional data that were obtained during the investigation of BST film on MgO(111) substrate. An explanation is given why these peaks are observed in the spectrum and an argument is given why they are not observed for other films in this work. Analysis of these peaks makes it possible to estimate the fraction of switched (modulated) polarization . Additionally, this makes it possible to estimate the values of the nonlinear susceptibilities, which is beyond the scope of this work.
4) The paper contains typos and mistakes, including in the formula of the material in which Ba is often misspelled with B.
These typos have been fixed.
Reviewer 3 Report
Report on the manuscript "Resonant excitation of the ferroelectric soft mode…" by K. Brekhov et al.
The authors fabricated three kinds of thin film samples of the BST system on the MgO(100), Si(001) and MgO(111) substrates, respectively. Next using a narrow-band THz pulse excitation they investigated the investigated the resonant excitation of the ferroelectric soft mode. It is proved that using the narrow-band THz pulse it is possible to effectively excite the ferroelectric soft mode in the investigated systems. I find the results interesting and originally enough to be published in "Nanomaterials". However, before the final decision, the manuscript calls for minor corrections:
1. All the experimental results like lattice parameters, thickness, etc. must be reported together with their uncertainties. Knowledge of the uncertainties of the results is of primary importance. For a proper data reporting, I recommend the authors browsing the ISO Guide to the Expression of uncertainty in Measurements. Results without uncertainties do not have any scientific meaning and should not be published in scientific journals.
2. Is it true that the lattice parameter c of the sample type II is ten times larger than the respective parameter of the sample type I? In case this is so one must comment such results.
3. The manuscript calls for professional linguistic correction. Some phrases sound funny. Instead of "dependencies" (line 153) one should use "dependences". Physical quantities are not in political relationship for example. Instead of "…the second degree…" (Line 183)better use "square". Instead of "monocrystal" there should be "single crystal".
4. The edition of symbols of physical quantities must be homogeneous. In equations these symbols are correctly edited as italic, while in the body text one finds them as normal. In fact this means different symbols. For example "m" stands for meter, while "m" traditionally is understood as symbol of mass.
Only after taking into account the above remarks, the manuscript can be recommended for publication.
no
Author Response
Dear Reviewer,
Thank you for your time and valuable comments to improve our manuscript.
In response to your questions, we can answer the following:
- All the experimental results like lattice parameters, thickness, etc. must be reported together with their uncertainties. Knowledge of the uncertainties of the results is of primary importance. For a proper data reporting, I recommend the authors browsing the ISO Guide to the Expression of uncertainty in Measurements. Results without uncertainties do not have any scientific meaning and should not be published in scientific journals.
According to [DOI: 10.1134/S1063783415080314], the measurement errors were 0.0003 and 0.0005 nm for the parameters c and a, respectively. The uncertainties were added to the text (lines 99, 104 and 107).
The text was added to the manuscript (lines 88-91 ):” The film thickness was measured using an MII-4 microinterferometer and also a Zeiss Supra-25 scanning electron microscope in the “cross-section” and AFM image of the end face of the etched film. From the height of the resulting step, the BST film thickness was determined”.
- Is it true that the lattice parameter c of the sample type II is ten times larger than the respective parameter of the sample type I? In case this is so one must comment such results.
Thank you very much for the comment. It's a typo. The text has been corrected.
- The manuscript calls for professional linguistic correction. Some phrases sound funny. Instead of "dependencies" (line 153) one should use "dependences". Physical quantities are not in political relationship for example. Instead of "…the second degree…" (Line 183) better use "square". Instead of "monocrystal» there should be "single crystal".
Thank you very much. All corrections have also been added to the text.
- The edition of symbols of physical quantities must be homogeneous. In equations these symbols are correctly edited as italic, while in the body text one finds them as normal. In fact this means different symbols. For example, "m" stands for meter, while "m" traditionally is understood as symbol of mass.
All symbols in the text body were made as italic.
Round 2
Reviewer 1 Report
References for ultrafast switching of polarization at least include Nature Photonics 11, 390 (2017), Nature Comm. 10:3951 (2019), Materialia 27: 101681(2023) together with short descriptions of their mechanics.
